# Shifting molecular localization by plasmonic coupling in a single-molecule mirage

Mario Raab[1], Carolin Vietz[1], Fernando Daniel Stefani[2,3], Guillermo Pedro Acuna[1] & Philip Tinnefeld[1]

Over the last decade, two fields have dominated the attention of sub-diffraction photonics research: plasmonics and fluorescence nanoscopy. Nanoscopy based on single-molecule localization offers a practical way to explore plasmonic interactions with nanometre resolution. However, this seemingly straightforward technique may retrieve false positional information. Here, we make use of the DNA origami technique to both control a nanometric separation between emitters and a gold nanoparticle, and as a platform for super-resolution imaging based on single-molecule localization. This enables a quantitative comparison between the position retrieved from single-molecule localization, the true position of the emitter and full-field simulations. We demonstrate that plasmonic coupling leads to shifted molecular localizations of up to 30 nm: a single-molecule mirage.

[1] Institute for Physical & Theoretical Chemistry, and Braunschweig Integrated Centre of Systems Biology (BRICS), and Laboratory for Emerging Nanometrology (LENA), Braunschweig University of Technology, Rebenring 56, 38106 Braunschweig, Germany. [2] Centro de Investigaciones en Bionanociencias (CIBION), Consejo Nacional de Investigaciones Científicas y Técnicas (CONICET), Godoy Cruz 2390, C1425FQD Ciudad de Buenos Aires, Argentina. [3] Departamento de Física, Facultad de Ciencias Exactas y Naturales, Universidad de Buenos Aires, Güiraldes 2620, C1428EAH Ciudad de Buenos Aires, Argentina. Correspondence and requests for materials should be addressed to F.D.S. (email: fernando.stefani@cibion.conicet.gov.ar) or to G.P.A. (email: g.acuna@tu-bs.de) or to P.T. (email: p.tinnefeld@tu-bs.de).

Plasmonics makes use of surface plasmon polaritons confined to nanometrically structured metals in order to control optical fields and interactions at the nanoscale[1,2]. Locally intensified optical fields around metallic nano particles have enabled highly sensitive detection of molecular fluorescence[3,4] and Raman scattering[5]. Also, plasmonic structures can produce strong modifications of the local photonic mode density, which in turn enables the control of rates[6–8] and directionality of molecular emission[9,10].

Outstanding experimental and theoretical efforts during the past decade have driven the transition of far-field fluorescence microscopes into nanoscopes capable of theoretically unlimited resolution while maintaining the simplicity and versatility of far-field optical systems relying on conventional lenses[11]. Different fluorescence nanoscopy methodologies have been developed, both in coordinate targeted versions, and based on stochastic localization of single molecules[12–14].

The combination of plasmonics as a platform for the control of nanoscale optical fields and interactions, along with far-field fluorescence nanoscopy as a versatile tool to locate single emitters and image sub-diffraction fields, is only emerging[15]. As such, these methods enable new experiments to study the interaction of single emitters with plasmonic structures[16–18].

Recently, several studies have applied localization based fluorescence nanoscopy near plasmonic nanostructures aiming to reveal nanometric details about optical near fields[16,17,19], nanoparticle geometry[20,21], surface chemistry[22] and electro-magnetic interactions with nearby molecules[23]. Interestingly, this seemingly straightforward approach does not always deliver accurate information of dye positions and electric field intensity distributions due to coupling of the emitters to localized surface plasmon resonance modes of the nanostructures. Examples include the different locations for luminescence and surface-enhanced Raman scattering (SERS) emission centres[18], mismatches between nanorod dimensions determined by super-resolution imaging and atomic force microscopy[21] and displaced localization of molecules close to metal nanoparticles[24], nanorods[22,25] and nanowires[19].

Far-field optical imaging of an isolated and point-like emitter yields a diffraction limited signal centred at the emitter position that enables its precise localization (Fig. 1a). Nanoscopy based on successive single-molecule localizations can deliver resolution below 10 nm[26]. However, if an emitter is located in the nanometric vicinity of a metallic nanoparticle, the near field of the emitter can induce currents in the nanoparticle and generate an image dipole that acts as a second source of radiation[27]. The relative probability of the two emission channels depends on the coupling of the emitter to the nanoparticle plasmon modes. The image obtained is the result of interference between these two sources of radiation, and corresponds neither to the emitter nor to the nanoparticle position (Fig. 1b). In spite of the fundamental and practical implications of this phenomenon for nano-photonics, a definite experimental verification with a quantitative comparison with full-field simulations has remained elusive owing to the inherent impossibility of optically measuring the true position of the molecular emitter[27,28].

In this work, we use a self-assembly scheme based on the DNA origami technique to fabricate hybrid nanosystems containing spherical gold nanoparticles (AuNPs) and fluorophores at controlled nanometric separations. The DNA origami provides a reference frame for the required *a priori* knowledge of the relative fluorophore position with respect to the AuNP without optical feedback[4,29–31]. In addition, it provides a platform for DNA-PAINT (points accumulation for imaging in nanoscale topography) super-resolution imaging of single molecules[32]. We combine 3D single-molecule localization nanoscopy and finite-difference frequency-domain (FDFD) full-field simulations to demonstrate that the far-field emission of single emitters is modulated by plasmonic coupling to AuNPs, leading to considerable mislocalizations in analogy to a single-molecule mirage.

## Results

**Direct observation of the plasmonic single-molecule mirage.** In an initial experiment, we used a 12-helix bundle (12HB) DNA origami that forms a rigid nanorod with a length of 228 nm and a diameter of 14 nm. The 12HB includes three marks for DNA-PAINT separated by a distance of 80 nm from each other and a docking site for a DNA-modified AuNP (Fig. 2a). Super-resolution fluorescence microscopy imaging of the 12HB yields three aligned spots separated by 80 nm, in accordance with the DNA origami design (Fig. 2b). Upon binding of an AuNP on the 12HB, the three localization spots do not appear in one line anymore (Fig. 2c). The central localization appears displaced from the line defined by the two extreme localizations, providing clear evidence for the single-molecule mirage.

Although this experiment demonstrates the single-molecule mirage induced by the AuNP, it is not adequate for a quantitative comparison to calculations because the deviation produced by the AuNP is three-dimensional (Fig. 2d, additional 3D-images in Supplementary Fig. 1), and the coupling of the fluorophores at the outer marks cannot be excluded.

**Quantitative analysis of the plasmonic single-molecule mirage.** For a quantitative determination, we designed an assay using two different structures based on a rectangular DNA origami, as illustrated in Fig. 3a,b. One structure (reference) contained three DNA-PAINT marks arranged in an equilateral triangle with 50 nm side length. The other structure (sample) is modified to have a single DNA-PAINT mark and a single docking site for a single AuNP. The three spots of the reference structure determine a plane from which axial deviations in the localization of the sample structures can be quantified (Fig. 3b). The reference structures are easily identified optically by their triangular

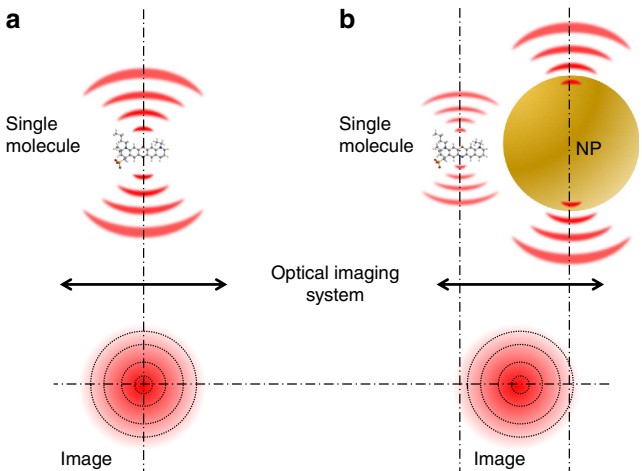

**Figure 1 | Schematic representation of the single-molecule mirage induced by a metallic nanoparticle (NP).** (**a**) The image of an isolated molecule is centred at the position of the molecule. (**b**) Imaging of a single molecule coupled to a metallic nanoparticle in the near-field leads to an image that may correspond neither to the position of the molecule nor to the position of the NP.

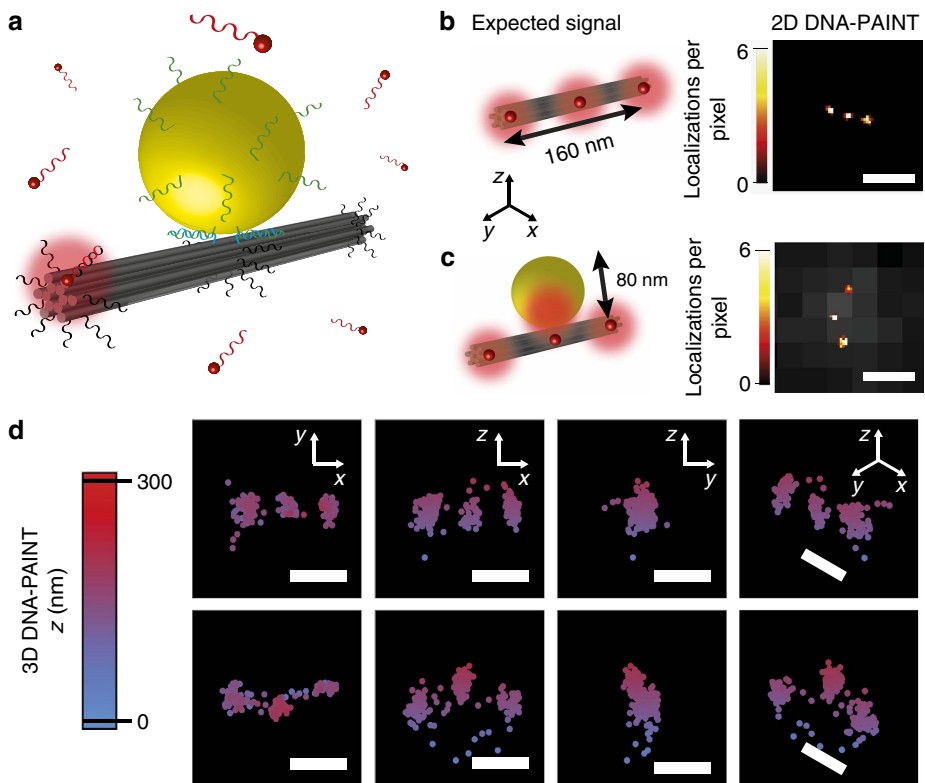

**Figure 2 | Observation of the single-molecule mirage.** (**a**) Working principle of the assay using DNA origami to observe the shift in molecular localization produced by plasmonic coupling. The 12-helix bundle (12HB) DNA origami structure provides sites for the dynamic binding of single fluorophores at three regions (at each end and at the centre) separated by 80 nm. In addition, at the centre of the 12HB there are docking sites for the incorporation of a DNA-modified AuNP in close contact (zipper configuration)[37], on top of the 12HB. The DNA origami structures are fixed on the substrate with random orientations. (**b,c**) Schematic depicting the expected emission spots next to a representative 2D DNA-PAINT image of a 12HB without (**b**) and with an 80 nm AuNP attached (**c**). The presence of the AuNP is evidenced by its scattering signal (overlaid in grey scale). (**d**) 3D DNA-PAINT imaging of a 12HB without (top) and with (bottom) an 80 nm AuNP attached. Scale bars: (**b,c**) 200 nm. (**d**) 100 nm.

geometry, whereas the sample structures including a NP are identified as the ones with a single emitter localization (inset of Fig. 3b, further details in Methods section). This assay enables the quantification of the localization displacement produced by AuNPs of different sizes. We investigated the effect of AuNPs with diameters of 20, 40, 60 and 80 nm. The expected distances between the DNA-PAINT markers and the AuNPs surface are included in Supplementary Table 1. In each case, multiple (~10,000) localizations on the reference and sample origami were acquired. An example of the distributions of axial localization ($z$) obtained from the reference and the sample structures with 80 nm AuNPs is shown in Fig. 3c; in this case, the presence of the AuNP leads to an average mislocalization of 29 nm.

We note that far-field imaging of dipolar emitters with a fixed orientation near a dielectric interface may present distorted signals and lead to systematic localization inaccuracies[33,34]. The same may occur with fixed dipolar emitters near metallic structures[28]. Dye molecules attached to DNA emit isotropically owing to fast rotation, but if they are strongly coupled to surface plasmons they show polarized emission[4]. The single-molecule signals in our experiments are nearly identical to the point-spread function of the optical system, showing no detectable distortion even for the larger AuNPs of 80 nm. We also confirmed the specificity of the detected structures used for further analysis (Supplementary Figs 2–5). Still, the polarized emission of the coupled molecules may have an influence on individual localizations. The magnitude and direction of the effect will depend on the orientation of the dye-NP system with respect to the imaging coordinates set by the orientation of the cylindrical lens. As the DNA origamis carrying the nanoparticles and dyes are randomly oriented on the surface, any orientational effect of individual structures cancels out in the average determined from a large number of structures. Therefore, any orientational dependency of the localization accuracy does not affect our determination of the average positions, but is certainly a contributing factor that broadens the distributions of $z$-positions (for example, Fig. 3c).

The effect of NP size on the axial mislocalization is summarized in Fig. 3d. As the NP diameter increases, the molecules appear higher in the $z$-direction, following an approximately linear trend. Extrapolation to zero diameter NPs shows a deviation from the linear trend for smaller sizes. In order to understand these results, we performed FDFD numerical simulations (details provided in the Methods section). The simulations considered spherical AuNPs with a size distribution according to a previous transmission electron microscopy characterisation (transmission electron microscopy images of used AuNPs can be found in ref. 4), and a range of separation distances between the fluorophores and the AuNPs that are, in principle, allowed by the DNA origami design (further details in Supplementary Table 1). Although fluorophores in the experiments can rotate freely, we considered a dipolar emitter oriented radially with respect to the NP based on previous results, which showed that the contribution of tangentially oriented dipoles should be neglected owing to strong quenching[4]. The simulated localizations computed at the

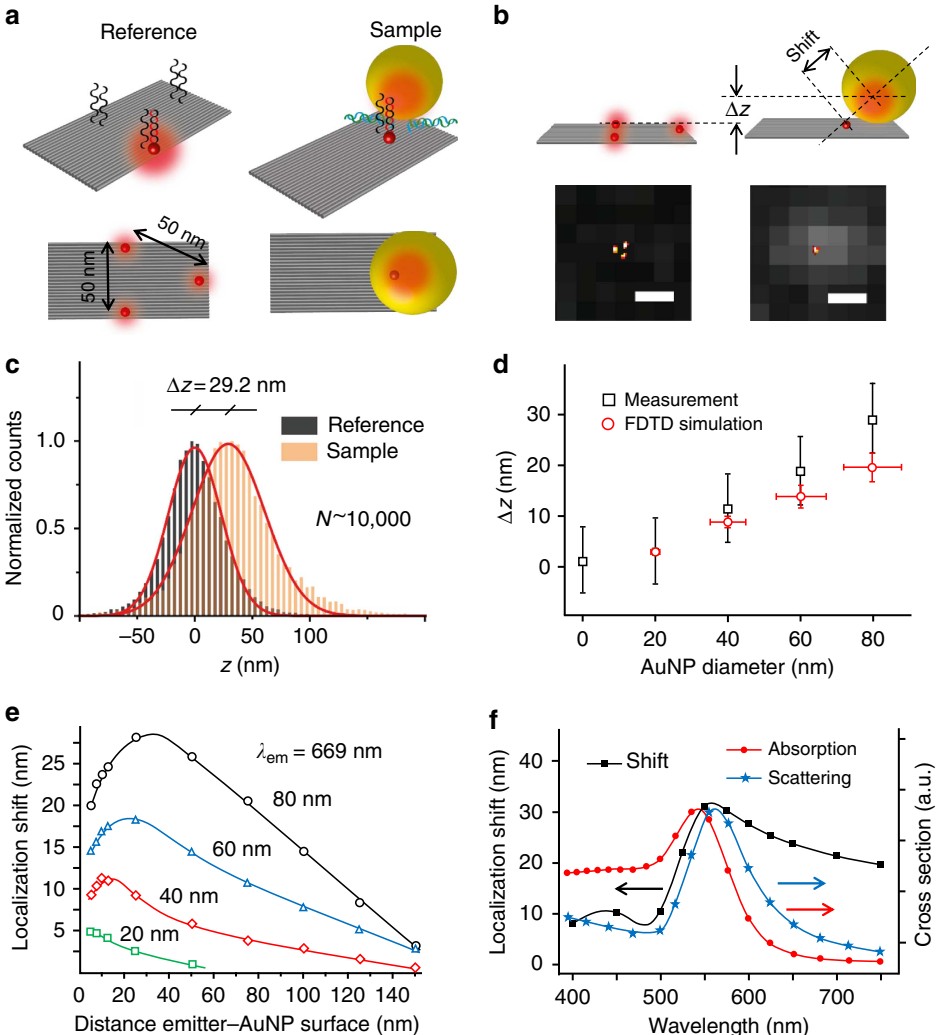

**Figure 3 | Quantification of the localization shift induced by gold nanoparticles (AuNPs).** (**a**) Schematic representation of the DNA origami structures used in the assay (perspective and top views). (**b**) Scheme showing the total shift in the position of the emission centre and the shift in axial position ($\Delta z$) induced by the AuNP, together with representative 2D DNA-PAINT super-resolution images of the reference and sample structures (scale bars, 200 nm). (**c**) Distributions of axial localizations obtained in the reference and the sample with 80 nm AuNPs. (**d**) Average shift produced by the AuNPs as a function of the AuNP diameter, together with the predicted shift obtained from full-field 3D simulations. Experimental error bars indicate $3\sigma$ in the average $\Delta z$ (details in Supplementary Note 1). The error bars in the simulations correspond to allowed dimensional ranges in the DNA origami structures (see Supplementary Table 1) and the size distributions of the AuNPs. (**e**) Simulated localization shift produced by AuNPs of different diameters, as a function of the separation distance between the emitter and the AuNP surface. (**f**) Simulated localization shift as a function of the emission wavelength for an emitter placed at 10 nm from an 80 nm AuNP, together with the scattering and the absorption spectra of the AuNP.

emission maximum wavelength of the fluorophore compare well to the experiments. Slightly smaller $z$-positions in the simulations compared with the experimentally determined average positions might be related to the influence of the local environment including DNA origami and single-stranded DNA on the optical properties, as was suggested recently[35].

In order to understand the extent that this localization error will influence experiments and applications, it is relevant to analyse the effect of other experimental variables on the mislocalization, such as the relative position of the emitter with respect to the AuNP and the emission wavelength. In Fig. 3e, the total shift in localization produced by AuNPs of different diameters, as a function of the separation distance from the emitter to the AuNP surface, is shown. The overall magnitude of the mirage shift increases with the AuNP size. As a function of the separation between the NP surface and the emitter, the mirage produced by the AuNPs is maximum for a distance smaller than

the AuNP radius. For example, the 80 nm AuNP maximum mislocalization occurs when the molecule is placed at $\sim 30$ nm from the AuNP surface. The effect reduces with increasing separation and, only for distances larger than 120 nm, becomes negligible in the context of super-resolution localization. Finally, we studied the mirage effect for different emission wavelengths, as shown in Fig. 3f. Because the mirage effect depends on the near-field coupling, the far-field absorption and scattering spectra do not provide a suitable measure. The magnitude of the mislocalization follows the same spectral dependence as the fluorescence enhancement[36], it is minimal at the blue side of the surface plasmon resonance, reaches a maximum close to the scattering maximum, and decays towards longer wavelengths at a much lower rate than the absorption or scattering cross-sections.

In summary, we have demonstrated that the near-field electromagnetic coupling of fluorophores to metallic

NPs leads to far-field images centred at positions that correspond neither to the fluorophore nor to the NP centre. In contrast to other reported experiments where mislocalizations were accompanied by drastic distortions of the single-molecule signals[19,28], the plasmon induced localization shifts we observed occur in the absence of any distortion of the single-molecule signals; that is, the signals of single molecules coupled to the AuNPs are identical to the point-spread function of the optical system. This single-molecule mirage effect can be modelled accurately using full-field numerical simulations. It depends strongly on the size and spectral properties of the plasmonic structure, the relative position and orientation of the emitter and the emitter's emission frequency. Our investigation using gold nanospheres represents the simplest expression of this phenomenon. Much larger modulations of the far-field emission are possible using other plasmonic nanostructures, such as nanorods. The use of single-molecule localization for the investigation of plasmonic fields and interactions should consider this effect. For example, accurate mapping of plasmonic near fields using super-resolution localization would be possible using two-photon excitation or up-conversion nanoparticles[36]. In this way, emission can be tuned to the blue side of the plasmon resonance, minimising the mislocalization. In addition, we believe that deeper understanding and control of this phenomenon will enable the use of plasmonic structures as near-field transducers for the far-field determination of orientations and/or emission frequencies of single emitters in non-scanning geometries. Compact near-field polarizing and dichroic devices can be envisaged, among other possibilities.

## Methods

**AuNP-functionalization.** AuNPs of 20–80 nm in diameter (BBI solutions) were functionalized with T20 single-stranded DNA-oligonucleotides modified with a thiol group on the 3′ end (Ella Biotech GmbH) designed to bind in the zipper geometry to the origami[37]. In brief, 2 ml of AuNP solution were stirred with 20 μl Tween20 (10%, Polysorbate20, Alfa Aesar), 20 μl of a potassium phosphate buffer (4:5 mixture of monobasic and dibasic potassium phosphate, Sigma Aldrich) and an excess of the oligonucleotide solution. After stirring overnight, the solution was heated to 40 °C and then submitted to the following salting process. Sodium chloride was added every 5 min over an hour with increasing amounts up to a concentration of 750 mM using PBS buffer containing 3.3 M sodium chloride. To purify the functionalized AuNPs from the excess of oligonucleotides, the mixture was diluted 1:1 with 1 × phosphate-buffered saline (PBS) containing 10 mM NaCl, 2.11 mM P8709, 2.89 mM P8584, 0.01% Tween20 and 1 mM ethylenediaminetetraacetic acid (spinning buffer) and spun down. The supernatant was pipetted out and the particle pellet was re-suspended in the buffer. This spinning process was repeated six times.

**DNA origami design.** For the 12HB, each of the three DNA-PAINT marks consist of ~12 docking strands with average distances of ~80 nm. The docking site for the AuNP consists of six polyA-docking strands colocalized with the DNA-PAINT mark in the middle. Eight biotins anchors are distributed over the whole origami.

For the rectangular DNA origami, each of the DNA-PAINT marks consists of six docking strands. The reference structure has three marks in the geometry of an equilateral triangle with ~50 nm side lengths. The sample structure has a single DNA-PAINT mark at a distance of ~11 nm from the AuNP docking site. This AuNP docking site consists of three polyA-docking strands. Both structures have six biotin anchors. For the $z = 0$ control sample we attached 10 Atto532-dyes for colocalization instead of AuNP docking sites.

Schemes of the exact caDNAno-designs (http://cadnano.org/) and tables with the corresponding DNA sequences can be found in the Supplementary Tables 2–5 and Supplementary Data files 1–4.

**Sample preparation.** The preparation of DNA origamis, substrates modified with BSA-Biotin-NeutrAvidin, and the binding of the DNA origamis on those substrates were carried out according to recently published protocols[38]. In brief, surfaces were incubated with SuperBlock (PBS) (Thermo Scientific) blocking buffer for 10 min to achieve additional surface passivation. Then the AuNP solution (AuNPs in spinning buffer) was applied to the surface (~4 h for 20 nm AuNPs, overnight for AuNPs > 20 nm). For AuNPs < 60 nm, the sample was subsequently incubated with a 20 nM solution of A15-Cy3b oligonucleotides (in PBS with 10 mM MgCl₂). Between each incubation step the surface was

washed three times with PBS containing 10 mM MgCl₂. After washing, measurements were carried out in imaging buffer (spinning buffer with 5 nM Atto655-imager strands and 10 mM MgCl₂). The concentrations of the AuNP solutions were determined using a UV/VIS Spectrophotometer (Nanodrop 2000, Thermo Scientific) at an optical path length of 1 mm. The used optical densities were ~0.08 at 525 nm for the 20 nm AuNPs, ~0.02 at 532 nm for the 40 nm AuNPs, ~0.03 at 540 nm for the 60 nm AuNPs and ~0.1 at 555 nm for the 80 nm AuNPs. A scheme of the sample with its different components can be found in Supplementary Fig. 6.

**Imaging.** DNA-PAINT measurements were carried out on a custom-built total internal reflection fluorescence (TIRF) microscope, based on an inverted microscope (IX71, Olympus) placed on an actively stabilized optical table (TS-300, JRS Scientific Instruments) and equipped with a nosepiece (IX2-NPS, Olympus) for drift suppression. Fluorescence excitation was at 644 nm with a 150 mW laser (iBeam smart, Toptica Photonics) spectrally filtered with a clean-up filter (Brightline HC 650/13, Semrock) and at 532 nm with a 1 W laser (MPB Communications) spectrally filtered with an optical interference filter (Z532/647x, Chroma). Both laser lines were coupled into the microscope with a dual-colour-beamsplitter (Dual Line zt532/640 rpc, AHF Analysentechnik) and focused on the backfocal plane of an oil-immersion objective (× 100, NA = 1.4, UPlanSApo, Olympus) aligned for TIRF illumination. An additional × 1.6 optical magnification lens is applied in the detection resulting in an effective pixel size of 100 nm. The fluorescence light is guided through a cylindrical lens ($f = 1,000$ mm) to gain 3D information and spectrally filtered with emission filters (ET 700/75, Chroma) and (BrightLine 582/75, AHF Analysentechnik). The filters were changed between sequential acquisitions of the different laser lines. Images were recorded by an electron multiplying charge-coupled device camera (Ixon X3 DU-897, Andor).

For DNA-PAINT measurements, 6,000 frames per super-resolution image were acquired at an excitation intensity of ~3.7 kW cm⁻², 100 ms integration time and an electron multiplying gain of 5. Super-resolution images were reconstructed by subsequent localization of single molecules via Gaussian fitting using Matlab routines. Analysis of the localization data was carried out with software packages written in LabView2011, partially based on the software computer-aided evaluation of origami-based standards[38]. The TIRF imaging of the NPs was done at the same integration time and electron multiplying gain. The 80 nm and 60 nm NPs were detected by scattering signals obtained with a laser excitation of ~2 kW cm⁻². The 40 and 20 nm NPs were labelled with Cy3B dyes and detected by fluorescence signals obtained with ~255 W cm⁻². Chromatic aberrations were corrected using TetraSpeck-Beads (100 nm, Invitrogen, T7279). Details on the calibration processes of the 3D DNA-PAINT measurements are provided in the Supplementary Figs 3 and 4.

**Simulations.** Numerical simulations were performed using an FDFD commercial software (CST STUDIO SUITE, Microwave module). The ATTO655 fluorophores were modelled by a current source oscillating at a frequency corresponding to the wavelength of maximum fluorescence emission and a total length of 0.1 nm. In order to estimate the shift in the apparent emission centre, first the far-field emission pattern arising from the current source is calculated. Based on those results, the emission phase centre is estimated over an appropriate solid angle corresponding to the numerical aperture of our objective. As expected, in the absence of AuNPs the current source and the phase centre position coincide. However, with a single AuNP in the vicinity of the current source, the phase centre position deviates from the current source position shifting toward the NP position. This shift is employed in Fig. 3d–f. All simulations were performed considering a medium with a relative permittivity of 1.77 to mimic the buffer conditions. Further details are described in Supplementary Table 1.

**Data availability.** All relevant data are available from the authors.

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

## Acknowledgements

We are grateful to Carsten Forthmann, Frank Demming, Juan Pablo Paz and Christian Schmiegelow for fruitful discussions. This work was funded by the Braunschweig International Graduate School of Metrology B-IGSM and the DFG Research Training Group GrK1952/1 'Metrology for Complex Nanosystems', by a starting grant (SiMBA, EU 261162) of the European Research Council (ERC) and by the Deutsche Forschungsgemeinschaft (DFG, AC 279/2-1 and TI 329/9-1). F.D.S. is grateful to the DFG for a Mercator Fellowship. C.V. is grateful for a scholarship of the Studienstiftung des deutschen Volkes. P.T. is grateful for the visiting professor program of CONICET.

## Author contributions

M.R. designed, planned, prepared and executed experiments, programmed analysis software, analysed and interpreted data and wrote the manuscript. C.V. executed and improved the NP-functionalization. G.P.A and F.D.S. designed experiments, performed simulations, interpreted data and wrote the manuscript. P.T. planned experiments, interpreted data, wrote the manuscript and supervised the project.

## Additional information

**Competing financial interests:** The authors declare no competing financial interests.

