## [Peer Review File · Nature Communications]

Reviewers' comments:

Reviewer #1 (Remarks to the Author):

In their manuscript "Single Molecule Mirage: shifted molecular localizations through plasmonic coupling", Raab et al evaluate the effect that plasmonic nanoparticles exert on single molecule localizations using self-assembled DNA origami structures.

The study is well-designed and the claims are (in most part) supported by solid data. While I find the study well-executed and the findings important for the field of plasmonics, I have doubts if it is of general broad interest for the interdisciplinary readership of Nature Communications. I would thus recommend the manuscript for submission to a more specialized journal.

It seems obvious to me, that a coupled system of dyes and nanoparticles leads to apparent single molecule emission "in-between" both entities. I would find it more fitting, if the authors would make it clear in the introduction, that this is an effect that would be expected, rather than presenting it as observed but thus far completely unaccounted for.

In a revised version, I'd suggest the authors to clarify a few points:

- Figure 3c and d: What is the distance of the emitter to the AuNP?
- Did the authors perform corresponding experiments to simulations in Fig. 3e
- What could explain the observed "higher" z-position in experiments compared to simulation the authors observed?

Reviewer #2 (Remarks to the Author):

The authors demonstrate the use of DNA origami to control positions of individual single fluorescence molecules and the relative position of gold nanoparticles in order to correlate between the position retrieved from single molecule localization and the true position of the emitter. They realized that plasmonic coupling of fluorescence emission into the gold nanoparticles leads to shift of molecular localization of up to 30 nm.

This is a nice piece of work. The noble part of this work is that they extract 3D information, while many other researches have focused only on 2D.

I have several questions and comments.

- (1) The position of Au nanoparticles relative to the 12-helix bundle DNA origami is not very clear (Fig. 2). The nanoparticles are exactly ON the bundle? Could you show a SEM image of the structure?
- (2) The geometry of DNA origami, especially the triangle one, is not clear (Fig. 3). Could you provide better figure?
- (3) The discussion in Fig 3 is relevant only when the Au nanoparticle is exactly at the middle of the triangle. How can the authors be sure about it?
- (4) The dispersity of size and shape is critical in this manuscript. Please provide TEM image of each Au nanoparticles.
- (5) Please show the PSF of each experiment. Slight distortion of PSF could occur with large NPs such as 80 nm in diameter, which could affect on the localization.
- (6) How did the authors extract the z-shift information from the simulation (Figure 3d). Please explain the detail of the simulation.

Reviewer #3 (Remarks to the Author):

The authors present a study of the impact of gold nanoparticles on the apparent image of single fluorescent molecules recorded with a wide-field microscope. The core point is that due to the electrodynamic near-field interaction between particle and fluorescent molecule, the angular distribution of radiation of the particle-dye system is considerably changed with respect to the molecule alone, which leads to a systematic shift of the center position of the molecule's image. This effect is physically similar to the well-known impact of a dielectric interface on the image of an inclined molecule. In both cases, one sees a strong modulation of the angular distribution of radiation of the emitting system due to near-field coupling of the emitter to an inhomogeneous dielectric or dielectric/metallic environment. It would be very recommendable to mention this physical connection and to cite the respective literature. As it stands, the authors convey the impression that they discovered some completely new never-heard-of physical effect, which is certainly not true. Besides that, the paper is written very clearly, and the strength of the work relies in the perfect control which the authors have on their sample preparation using DNA origami. This allows them to position a fluorescence molecule with extremely well-controlled accuracy close to gold particles of perfectly known size, which makes a quantitative comparison between experiment and theory possible. Unsurprisingly, the presented theoretical calculations using Maxwell's electrodynamics confirm the measured effect.

Technische Universität Braunschweig | Physikalische und
Theoretische Chemie | Hans-Sommer-Str. 10 | 38106 Braunschweig |

Technische Universität Braunschweig
**Institut für Physikalische und
Theoretische Chemie**
Abt. NanoBioSciences
Hans-Sommer-Str. 10
38106 Braunschweig
Prof. Dr. Philip Tinnefeld
Tel. +49 (0) 531 391-5330 / 5342
Fax +49 (0) 531 391-7305
p.tinnefeld@tu-braunschweig.de
www.tu-
braunschweig.de/pci/forschung/tinnefeld

Braunschweig, 15.09.2016

Reviewers' comments:

Reviewer #1 (Remarks to the Author):

In their manuscript "Single Molecule Mirage: shifted molecular localizations through plasmonic coupling", Raab et al evaluate the effect that plasmonic nanoparticles exert on single molecule localizations using self-assembled DNA origami structures.

The study is well-designed and the claims are (in most part) supported by solid data. While I find the study well-executed and the findings important for the field of plasmonics, I have doubts if it is of general broad interest for the interdisciplinary readership of Nature Communications. I would thus recommend the manuscript for submission to a more specialized journal.

It seems obvious to me, that a coupled system of dyes and nanoparticles leads to apparent single molecule emission "in-between" both entities. I would find it more fitting, if the authors would make it clear in the introduction, that this is an effect that would be expected, rather than presenting it as observed but thus far completely unaccounted for.

We are grateful to reviewer for pointing this impression out. It was not our intention to claim that our observations were unexpected. We think that the following sentences of our introduction state quite clearly that we are dealing with a known phenomenon, and that we provide a, so far elusive, quantitative experimental verification:

"However, if an emitter is located in the nanometric vicinity of a metallic nanoparticle, the near-field of the emitter can induce currents in the nanoparticle, generating an image dipole that acts as a second source of radiation²⁶"

"In spite of the fundamental and practical implications of this phenomenon for nano-photonics, a definite experimental verification with a quantitative comparison to full-field simulations has remained elusive due to the inherent impossibility of measuring optically the true position of the molecular emitter^{26,28}"

Nevertheless, we believe that the passage of the introduction that has accidentally conveyed the impression of presenting unexpected results was the following:

“Interestingly, this seemingly straightforward approach produced controversial data, such as...”

We have modified it so as to mention the plasmon-emitter coupling as responsible for the observed effects:

“Interestingly, this seemingly straightforward approach does not always deliver accurate information of dye positions and electric field intensity distributions due to coupling of the emitters to the localized surface plasmon resonance modes of the nanostructures. For example, ...”

In a revised version, I'd suggest the authors to clarify a few points:

- Figure 3c and d: What is the distance of the emitter to the AuNP?

We have included the following table in the SI.

Table 1. Mean NP-fluorophore distances

NP-diameter (nm)	mean NP-fluorophore distance (nm)
20	9.1
40	7.7
60	7.0
80	6.6

And the corresponding reference on the main text:

“The distances between the DNA-PAINT markers and the AuNPs surface are included in table 1 of the SI.”

- Did the authors perform corresponding experiments to simulations in Fig. 3e

We did not perform such experiments. Nonetheless, given the satisfactory agreement between our experimental results and the numerical simulations, we consider that those numerical results are a useful guideline for future experiments.

- What could explain the observed “higher” z-position in experiments compared to simulation the authors observed?

There must be a parameter of the calculation that is not fully accurate. One possibility is that we have not included the DNA origami in the simulations. In a recent report:

Nature Communications 5, 3448 (2014). doi:10.1038/ncomms4448

it is shown that numerical simulations reproduce correctly the scattering spectra of gold nanoparticle dimers including DNA origami, only if the DNA was considered and actually, with a surprisingly high refractive index value.

We have therefore added the following line:

“Slightly smaller z-positions in the simulations compared to the experimental average positions might be related to the influence of the local environment including DNA origami and single stranded DNA on the optical properties as was suggested recently³⁶”

Reviewer #2 (Remarks to the Author):

The authors demonstrate the use of DNA origami to control positions of individual single fluorescence molecules and the relative position of gold nanoparticles in order to correlate between the position retrieved from single molecule localization and the true position of the emitter. They realized that plasmonic coupling of fluorescence emission into the gold nanoparticles leads to shift of molecular localization of up to 30 nm.

This is a nice piece of work. The noble part of this work is that they extract 3D information, while many other researches have focused only on 2D.

We thank the reviewer for thorough revision, positive remark on the quality and novelty of our manuscript. His/her comments gave us the chance to further clarify relevant details of our experiments and helped improving the manuscript.

I have several questions and comments.

- (1) The position of Au nanoparticles relative to the 12-helix bundle DNA origami is not very clear (Fig. 2).*
- 2). The nanoparticles are exactly ON the bundle? Could you show a SEM image of the structure?*

The AuNPs are tightly bound to the 12HB on the opposite side of the biotin strands. Due to linker lengths and surface roughness, the nanoparticle-12HB constructs can adopt a range of possible orientations with respect to the substrate normal. We have modified the caption of Figure 2 to make this clearer.

Unfortunately, our sample preparation method is not compatible with electron microscopy imaging. In order to prevent the formation of DNA origami-NP-aggregates, we first immobilize the origami structures on a glass slide which was previously functionalized with BSA-Biotin. Then the sample is incubated with the AuNPs, as described in the Methods part. We have not succeeded in obtaining SEM images showing clearly the DNA and the AuNPs when lying on BSA.

- (2) The geometry of DNA origami, especially the triangle one, is not clear (Fig. 3). Could you provide better figure?*

In the revised manuscript we provide an improved Figure 3a, including top-views of the DNA structures, which aid for a better visualization of the structures.

- (3) The discussion in Fig 3 is relevant only when the Au nanoparticle is exactly at the middle of the triangle. How can the authors be sure about it?*

This seems to be a misunderstanding: Only the reference structure has three DNA-PAINT markers forming a triangle in order to determine the reference z position. The sample structure has only a single DNA-PAINT marker near the NP. We believe that our modified figure 3 provides a more precise picture of both the reference and the sample structure.

(4) The dispersity of size and shape is critical in this manuscript. Please provide TEM image of each Au nanoparticles.

We agree with the reviewer that the size and shape of the AuNPs have critical influence on our measurements. We have verified that the AuNPs are nearly spherical, and have taken into account their size distribution in our calculations, as mentioned in the manuscript. In the revised version of our manuscript we have included a reference to a previous publication in which TEM images of the same AuNPs are shown:

“The simulations considered spherical AuNPs with a size distribution according to a previous TEM characterization (TEM images of used AuNPs can be found in the SI of reference ⁴)”

(5) Please show the PSF of each experiment. Slight distortion of PSF could occur with large NPs such as 80 nm in diameter, which could affect on the localization.

We agree that larger metallic nanostructures could induce distortions to the PSF as was recently found for more complex nanowires (see ref. 28). For our highly symmetric nanoparticles we do not observe such distortions as mentioned in the manuscript: “We remark that this effect occurs in the absence of any observable distortion of the single molecule signals; i.e. the signals of single molecules coupled to the AuNPs are identical to the point-spread function of the optical system.”

As the 3D information is achieved by the astigmatism method, the PSF is systematically distorted in x and y depending on the z-position, according to the direction set by the cylindrical lens. Any other distortion of the PSF due to the NPs would be in random directions, and therefore might contribute to the spread of detected z-positions but should not yield a shifted average. In the revised manuscript, we added a few sentences discussing this issue on page 7, as a part of the extra paragraph also motivated by Referee 3.

Nevertheless, motivated by the referee’s comment we have included a new figure S3 together with its corresponding discussion in which we give more details on the 3D measurements and provide exemplary PSFs for the measurements in the presence of 80 nm AuNPs as well as without nanoparticles. This gives the reader an additional impression of how the raw data looks, how the 3D information was obtained and that the nanoparticles do not distort the PSFs in any systematic way.

(6) How did the authors extract the z-shift information from the simulation (Figure 3d). Please explain the detail of the simulation.

We have extended the explanation of the Methods section by including the following lines:

“In order to estimate the shift in the apparent emission center, first the far field emission pattern arising from the current source is calculated. Based on those results, the emission phase center is estimated over an appropriate solid angle corresponding to the numerical aperture of our objective.. As expected,

in the absence of AuNPs, the current source and the phase center position coincide. However, with a single AuNP in the vicinity of the current source, the phase center position deviates from the current source position shifting toward the NP position. This shift is employed in figures 3 d-f..”

Reviewer #3 (Remarks to the Author):

The authors present a study of the impact of gold nanoparticles on the apparent image of single fluorescent molecules recorded with a wide-field microscope. The core point is that due to the electrodynamic near-field interaction between particle and fluorescent molecule, the angular distribution of radiation of the particle-dye system is considerably changed with respect to the molecule alone, which leads to a systematic shift of the center position of the molecule's image. This effect is physically similar to the well-known impact of a dielectric interface on the image of an inclined molecule. In both cases, one sees a strong modulation of the angular distribution of radiation of the emitting system due to near-field coupling of the emitter to an inhomogeneous dielectric or dielectric/metallic environment. It would be very recommendable to mention this physical connection and to cite the respective literature. As it stands, the authors convey the impression that they discovered some completely new never-heard-of physical effect, which is certainly not true. Besides that, the paper is written very clearly, and the strength of the work relies in the perfect control which the authors have on their sample preparation using DNA origami. This allows them to position a fluorescence molecule with extremely well-controlled accuracy close to gold particles of perfectly know size, which makes a quantitative comparison between experiment and theory possible. Unsurprisingly, the presented theoretical calculations using Maxwell's electrodynamics confirm the measured effect.

We thank the reviewer for the positive comments. Also, we agree with the reviewer that our work is related to the influence of a dielectric interface on the image of an inclined molecule. Furthermore, we believe that this orientational effect is an important contribution to the broadness of the z-position distribution we observe. As a matter of fact, there was a section in the SI dedicated to this issue, including references. In our revised submission, we have moved that information from the SI to the manuscript text, including the corresponding citations.

REVIEWERS' COMMENTS:

Reviewer #1 (Remarks to the Author):

I thank the authors for the clarifications made regarding my points. I think everything taken together, the manuscript can now be considered for publication at Nature Communications from my point of view.

Reviewer #2 (Remarks to the Author):

The authors unfolded all the reviewer's questions. The figures and manuscript is now very clear. Again, I think this is very careful and precise, and thus a nice work. I believe that the manuscript is now ready to be published.

Reviewer #3 (Remarks to the Author):

The authors have appropriately answered all questions and comments.